# An Analytical Model for the Excavation Damage Zone in Tunnel Surrounding Rock

Xiaoding Xu [1], Yuejin Zhou [1,*], Chun Zhu [2], Chunlin Zeng [1,*] and Chong Guo [1]

1   State Key Laboratory for Geomechanics & Deep Underground Engineering Xuzhou, China University of Mining & Technology, Xuzhou 221116, China
2   School of Earth Sciences and Engineering, Hohai University, Nanjing 210098, China
*   Correspondence: ts16030036a3@cumt.edu.cn (Y.Z.); zengcl@cumt.edu.cn (C.Z.)

**Abstract:** An accurate theoretical model to predict the extent and mechanical behavior of the excavation damage zone (EDZ) in the surrounding rock of deep-buried tunnel is of great importance for the practical engineering applications. Using the elastic-plastic theory and combined with the analysis on the stress characteristics of the tunnel surrounding rock, this paper present a predict model for the EDZ formation and evolution. A three-zone composite mechanical model was established for the tunnel surround rock and the corresponding stress state and displacement field of three zones were derived. The effects of the strain softening and dilatancy during rock deformation was taken into account. The correctness of the proposed model was validated by the existing theoretical models. A sensitivity analysis for different influencing factors in this model was also performed. The results can benefit for the future numerical and experimental studies.

**Keywords:** tunnel; excavation damage zone; elastic-plastic mechanics; dilatancy; strain softening





## 1. Introduction

Due to the influence of deep-buried tunnel excavation and high in-situ stress, the hydraulic, thermal, and mechanical properties of the surrounding rock around the excavation face is altered significantly [1], leading to the formation of the excavation damage zone (EDZ) or disturbed rock zone (DRZ) [2–6]. An accurate determination of its extent and mechanical behavior is of great importance for the design and construction of the deep underground engineering.

EDZ has been extensively studied experimentally and numerically. For example, by measuring the ultrasonic wave velocity and acoustic emission [7,8] of the surrounding rock, Falls and Young [9], Meglis et al. [10], and Martino and Chandler et al. [11] determined the formation and development of EDZ during the excavation phase. Even though their real-time performance and high accuracy, experiments cannot be widely applied during the entire design and construction process because they are high-cost and time-consuming. The numerical methods can well complement the experimental approaches. For example, Li and Liu [3] used a two-part Hooke's model implemented into FLAC3D [12] and simulated the EDZ formation and evolution around the ED-B tunnel at the Mont Terri site. The simulation results were found to be highly consistent with those from field tests. Considering the blasting-induced damage, Yang et al. [4] adopted LS-DYNA [13] and simulated the EDZ evolution process along with a deep-buried tunnel excavation by drill and blast. Zhu and Bruhns [14] adopted RFPA2D [15] to model the EDZ of a circular tunnel under a wide range of hydromechanical conditions and the concurrent microdamage evolution was captured. However, due to the complexity of the mechanical behaviour and nonlinear deformation characteristics of the EDZ, it is difficult for the pre-existing numerical models to take into considerations the time dependent dynamic formation and evolution of the EDZ induced by excavation process. And the numerical models are still time-consuming, largely limiting their practical applications.

Therefore, an accurate and easy-to-use predicable model is needed to provides convenience for extensive practical engineering design and safety assessment. Some analytical solutions have been proposed for the evaluation of the EDZ development. For example, Schartz and Einstein [16] established an analytical model based on the stiffness ratio between the tunnel support and the surrounding rock. Li and Wang [17] derived an analytical solution for the stress state and deformation field of the supported circular tunnel based on elastic theories and plane strain conditions. Considering the creep effect and the delay installation of the support, Fahimifar et al. [18] proposed the analytical solution to predict the time dependent deformation of the tunnel wall. However, the existing theoretical models are still relatively limited and incapable to comprehensively predict the extent and mechanical behavior of the EDZ, reflecting the omission of some key factors in the mechanical analysis.

To address the current research gap, in this paper, based on elastic-plastic mechanics, a novel analytical model for the EDZ in tunnel surrounding rock was established by further considering the effects of the strain-softening and dilatancy during the surrounding rock deformation. A sensitivity analysis is also performed to investigate the influence of various factors in the model.

## 2. Mechanical Division of Tunnel Surrounding Rock

The excavation of the tunnel by drilling and blasting destroys the original stress state of the surrounding rock, and the phenomenon of stress concentration occurs [19–21], which leads to the destruction of surrounding rock to form a crushed zone, a plastic softening zone, and an elastic zone. Here, we defined that the range of the EDZ includes the crushed zone and plastic zone. In addition, the lateral pressure coefficient will change as well when the tunnel is under the influence of mining. According to the above analysis, considering the influence factors such as lateral pressure coefficient and support resistance, the elastic-plastic mechanical calculation model of the tunnel surrounding rock was established, as shown in Figure 1.

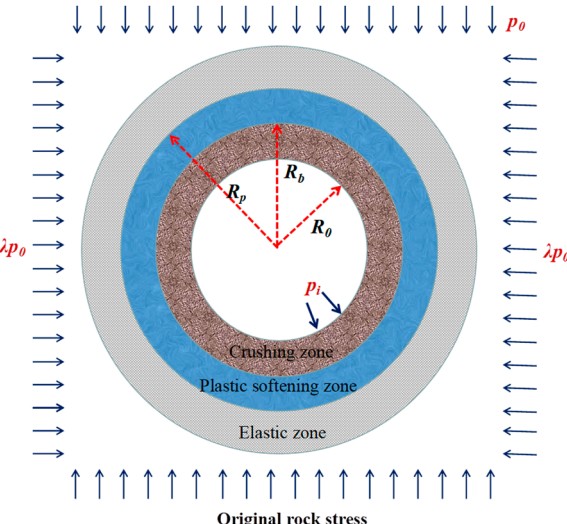

**Figure 1.** Mechanical model of tunnel surrounding rock.

As shown in Figure 1, assuming that the surrounding rock is a continuous homogeneous isotropic medium, the original rock stress at the infinite distance of the tunnel is $p_0$, and the lateral pressure coefficient is $\lambda$. After the tunnel excavation and stress redistribution, the fracture zone, the plastic softening zone and the elastic zone are formed outwardly in the surrounding rock successively. Suppose that the tunnel radius is $R_0$, the external radius of the crushed zone is $R_b$, the external radius of the plastic softening zone is $R_p$, and

the support resistance of the tunnel is $p_i$. The elastic theory gives the basic formulas as below [22]:

1. Yield criterion

$$\sigma_\theta = \frac{1 + \sin \varphi}{1 - \sin \varphi} \sigma_r + \frac{2c \cos \varphi}{1 - \sin \varphi} \tag{1}$$

where $\sigma_r$ is radial stress, $\sigma_\theta$ is tangential stress, $c$ is cohesion, and $\varphi$ is internal friction angle.

2. Balanced differential formula

$$\frac{d\sigma_r}{dr} + \frac{\sigma_r - \sigma_\theta}{r} = 0 \tag{2}$$

3. Geometric formula

$$\begin{cases} \varepsilon_r = \frac{du}{dr} \\ \varepsilon_\theta = \frac{u}{r} \end{cases} \tag{3}$$

where: $\varepsilon_r$ is the radial strain and $\varepsilon_\theta$ the tangential strain, and $u$ denotes radial displacement of surrounding rock.

4. Constitutive formula

$$\begin{cases} \varepsilon_r = \frac{1-\mu^2}{E} \left( \sigma_r - \frac{\mu}{1-\mu} \sigma_\theta \right) \\ \varepsilon_\theta = \frac{1-\mu^2}{E} \left( \sigma_\theta - \frac{\mu}{1-\mu} \sigma_r \right) \end{cases} \tag{4}$$

## 3. Mechanical Analysis of Three-Zone Composite Model

### 3.1. Mechanical Analysis of Elastic Zone

3.1.1. Stress Analysis of Elastic Zone

According to the stress characteristics of elastic zone in the surrounding rock, an equivalent model of elastic zone can be established, as shown in Figure 2. The surrounding rock in the elastic zone can be simplified as follows: the vertical pressure is $p_0$, the lateral pressure is $\lambda p_0$, the internal pressure is $\sigma_{ep}$, and the radial stress at the elastic-plastic interface is $\sigma_{ep}$. By using Kirsch's solution [23], the stress distribution in the elastic zone can be obtained as below:

$$\begin{cases} \sigma_r^e = \frac{1+\lambda}{2} p_0 \left( 1 - \frac{R_p^2}{r^2} \right) + \sigma_{ep} \frac{R_p^2}{r^2} - \frac{1-\lambda}{2} p_0 \left( 1 - 4\frac{R_p^2}{r^2} + 3\frac{R_p^4}{r^4} \right) \cos 2\theta \\ \sigma_\theta^e = \frac{1+\lambda}{2} p_0 \left( 1 + \frac{R_p^2}{r^2} \right) - \sigma_{ep} \frac{R_p^2}{r^2} + \frac{1-\lambda}{2} p_0 \left( 1 + 3\frac{R_p^4}{r^4} \right) \cos 2\theta \end{cases} \tag{5}$$

where $\sigma_r^e$ denotes the radial stress of elastic zone, $\sigma_\theta^e$ is the tangential stress of elastic zone, $R_p$ stands for the radius of plastic zone, and $\sigma_{ep}$ is the radial stress at the elastic-plastic interface.

When $\lambda = 1$, the stress of elastic zone can be calculated through Equation (5) with no consideration for the influence of lateral pressure coefficient.

When $r = R_p$, at the elastic-plastic interface of surrounding rock, there is:

$$\begin{cases} \sigma_r^{ep} = \sigma_{ep} \\ \sigma_\theta^{ep} = (1 + \lambda) p_0 - \sigma_{ep} + 2(1 - \lambda) p_0 \cos 2\theta \end{cases} \tag{6}$$

At the elastic-plastic interface, the stress also satisfies the Mohr-Coulomb criterion, from which the following can be obtained:

$$\sigma_\theta^{ep} = (1 + \lambda) p_0 - \sigma_{ep} + 2(1 - \lambda) p_0 \cos 2\theta = \frac{1 + \sin \varphi}{1 - \sin \varphi} \sigma_{ep} + \frac{2c \cos \varphi}{1 - \sin \varphi} \tag{7}$$

Solving Equation (7) gives:

$$\sigma_{ep} = (1 - \sin\varphi)\left[\frac{1+\lambda}{2}p_0 + (1-\lambda)p_0\cos 2\theta\right] - c\cos\varphi \qquad (8)$$

Substitute Equation (8) into Equation (6) and the following can be obtained:

$$\begin{cases} \sigma_r{}^{ep} = (1 - \sin\varphi)\left[\frac{(1+\lambda)}{2}p_0 + (1-\lambda)p_0\cos 2\theta\right] - c\cos\varphi \\ \sigma_\theta{}^{ep} = \frac{3}{4}p_0(1+\lambda) + \frac{5}{2}p_0(1-\lambda)\cos 2\theta + c\cos\varphi \end{cases} \qquad (9)$$

It can be seen from Equation (9) that, considering the influence of lateral pressure coefficient, the stress state of the elastic zone is not only related to the original rock stress, but also to the lateral pressure coefficient and angle. In other words, the tunnel stress state varies at different angles. The lateral pressure coefficient has a great influence on the distribution of stress field.

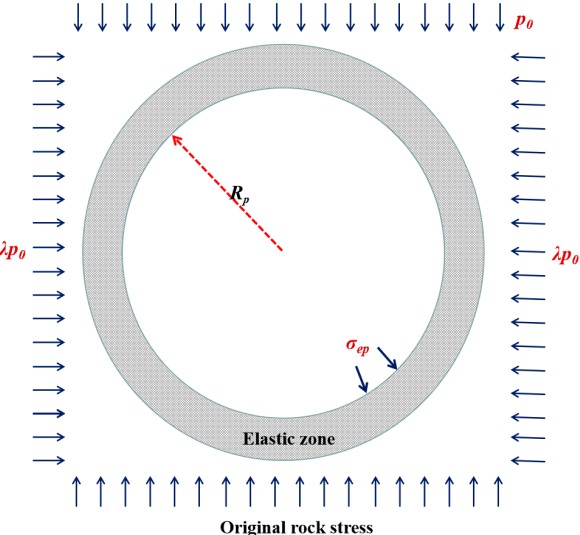

**Figure 2.** Equivalent model of elastic zone for tunnel surrounding rock.

### 3.1.2. Displacement Analysis of Elastic Zone

By substituting the stress results into the constitutive Equation (4), the strain of the elastic zone can be obtained as follows:

$$\varepsilon_\theta = \frac{1+\mu}{E}\left\{ \begin{array}{l} \frac{1+\lambda}{2}p_0\left[(1-2\mu) + \frac{R_p{}^2}{r^2}\right] - \sigma_{ep}\frac{R_p{}^2}{r^2} + \\ \frac{1-\lambda}{2}p_0\cos 2\theta\left[1 - 4\mu\frac{R_p{}^2}{r^2} + 3\frac{R_p{}^4}{r^4}\right] \end{array} \right\} \qquad (10)$$

By substituting Equation (10) into geometric Equation (3), the displacement of elastic region can be obtained as follows:

$$u_e = \frac{1+\mu}{E}\left\{ \begin{array}{l} \frac{1+\lambda}{2}p_0\left[(1-2\mu)r + \frac{R_p{}^2}{r}\right] - \sigma_{ep}\frac{R_p{}^2}{r} + \\ \frac{1-\lambda}{2}rp_0\cos 2\theta\left[1 - 4\mu\frac{R_p{}^2}{r^2} + 3\frac{R_p{}^4}{r^4}\right] \end{array} \right\} \qquad (11)$$

### 3.2. Mechanical Analysis of Plastic Zone

When the external load on the surrounding rock exceeds its strength, the cohesion and internal friction angle of surrounding rock will reduce to varying degrees. This process is known as strain softening. Therefore, the strain softening effect on surrounding rock

must be considered in mechanical analysis. For the convenience of calculation, the strain softening model adopted in this paper is shown in Figure 3, in which the cohesive strain softening model is a three line model. Since the internal friction angle changes in a small range and has little influence on the stress distribution, it is assumed that the internal friction angle of rocks in elastic and plastic zones is $\varphi_0$, and that in crushed zone is $\varphi_b$.

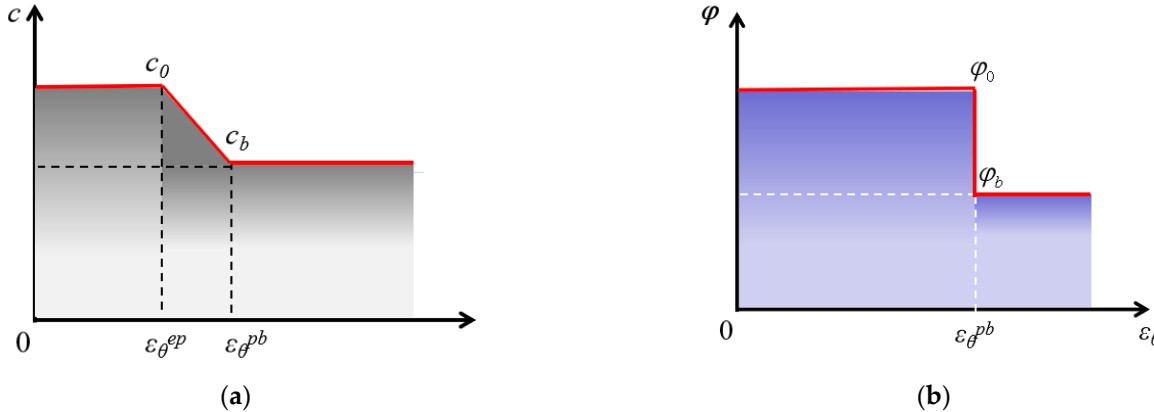

**Figure 3.** Strain-softening model of tunnel surrounding rock. (**a**) Cohesion strain-softening model; (**b**) Internal friction angle strain-softening model.

According to Figure 3, the softening modulus of cohesion could be expressed as:

$$M_c = \frac{c_0 - c_b}{\varepsilon_\theta^{pb} - \varepsilon_\theta^{ep}} \tag{12}$$

where: $c_0$ denotes the initial cohesion, $c_b$ is the residual cohesion, $\varepsilon_\theta^{ep}$ stands for the critical tangential strain at the interface of elastic zone and plastic softening zone, $\varepsilon_\theta^{pb}$ refers to the critical tangential strain of surrounding rock at the interface between the plastic softening zone and the crushed zone.

The surrounding rock of the plastic softening zone is assumed to be incompressible, and $\varepsilon_\theta + \varepsilon_z + \varepsilon_r = 0$. The plane strain condition is $\varepsilon_z = 0$ and $\varepsilon_\theta + \varepsilon_r = 0$, so the geometric Equation (3) is expressed as follows:

$$\frac{u}{r} + \frac{du}{dr} = 0 \tag{13}$$

To solve the differential Equation (13), we can get $u = \frac{C_1}{r}$, where $C_1$ is the integral constant, which is related to the boundary conditions.

From the definition of equivalent strain, the equivalent effect of surrounding rock in the plastic softening zone evolves to be:

$$\varepsilon_i = \frac{\sqrt{2}}{3}\sqrt{(\varepsilon_\theta - \varepsilon_r)^2 + (\varepsilon_r - \varepsilon_z)^2 + (\varepsilon_z - \varepsilon_\theta)^2} = \frac{2\sqrt{3}}{3}\varepsilon_\theta = \frac{2\sqrt{3}}{3}\frac{C_1}{r^2} \tag{14}$$

When $r = R_p$, and $\varepsilon_i = \varepsilon_\theta^{ep}$, with the contact conditions of the interface between the elastic zone and the damage zone taken into consideration, the following equation can be obtained:

$$C_1 = \frac{\sqrt{3}}{2}\varepsilon_\theta^{ep}R_p^2 \tag{15}$$

When Equation (15) is substituted into Equation (14), it can be concluded that the effect in the softening zone is as follows:

$$\varepsilon_i = \frac{2\sqrt{3}}{3}\varepsilon_\theta = \left(\frac{R_p}{r}\right)^2\varepsilon_\theta^{ep} \tag{16}$$

The tangential strain of the plastic softening zone is:

$$\varepsilon_\theta = \frac{\sqrt{3}}{2}\left(\frac{R_p}{r}\right)^2 \varepsilon_\theta{}^{ep} \tag{17}$$

The relationship between the cohesion and the radius after the surrounding rock softening can be deduced as follows:

$$c^* = c_0 - M_c \varepsilon_\theta{}^{ep}\left(\frac{\sqrt{3}}{2}\left(\frac{R_p}{r}\right)^2 - 1\right) \tag{18}$$

At the same time, the dilatancy effect of surrounding rock at yield is considered in the plastic zone as well. The model of dilatancy effect of surrounding rock is shown in Figure 4, where $\eta_1$ and $\eta_2$ denote the dilatancy gradients of plastic softening zone and crushed zone of surrounding rock respectively, which are assumed to be constant.

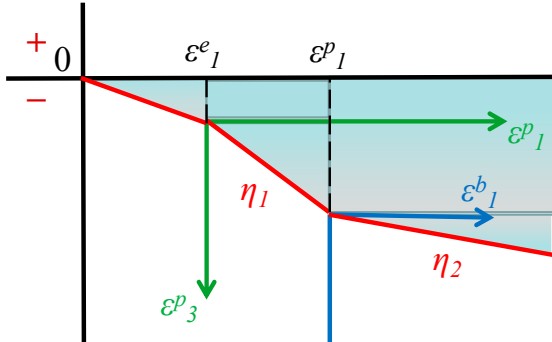

**Figure 4.** Dilatation effect model of tunnel surrounding rock.

In the plastic softening zone, the plastic flow rule is as follows:

$$\varepsilon_r{}^p + \eta_1 \varepsilon_\theta{}^p = 0 \tag{19}$$

In the crushed zone, the plastic flow rule is as follows:

$$\varepsilon_r{}^b + \eta_2 \varepsilon_\theta{}^b = 0 \tag{20}$$

### 3.2.1. Stress of Plastic Softening Zone

According to Equations (1) and (18), the yield criterion of plastic softening zone is as follows:

$$\sigma_\theta{}^p = n\sigma_r{}^p + mc^* \tag{21}$$

where: $n = \frac{1+\sin\varphi_0}{1-\sin\varphi_0}$, $m = \frac{2\cos\varphi_0}{1-\sin\varphi_0}$.

By substituting Equation (21) into Equation (2), it can be deduced that:

$$\frac{d\sigma_r{}^p}{dr} + \frac{1-n}{r}\sigma_r{}^p = \frac{mc^*}{r} \tag{22}$$

To solve Equation (22), we can get:

$$\sigma_r{}^p = C_2 r^{n-1} + m\left[\frac{c_0 + M_c\varepsilon_\theta{}^{ep}}{1-n} + \frac{\frac{\sqrt{3}}{2}M_c\varepsilon_\theta{}^{ep}}{1+n}\left(\frac{R_p}{r}\right)^2\right] \tag{23}$$

Combined with boundary conditions: $r = R_p$, $\sigma_r = \sigma_{ep}$, the following can be obtained:

$$C_2 = \frac{\sigma_{ep} - m(A+B)}{R_p{}^{n-1}} \tag{24}$$

where: $A = \frac{c_0 + M_c \varepsilon_\theta{}^{ep}}{1-n}$, $B = \frac{\sqrt{3}}{2} \frac{M_c \varepsilon_\theta{}^{ep}}{1+n}$.

By substituting Equations (23) and (24) into Equation (21), the distribution of stress field in the plastic softening zone can be obtained:

$$\begin{cases} \sigma_r{}^p = [\sigma_{ep} - m(A+B)] \left(\frac{R_p}{r}\right)^{1-n} + m\left[A + B\left(\frac{R_p}{r}\right)^2\right] \\ \sigma_\theta{}^p = n\sigma_r + m\left[c_0 - \frac{\sqrt{3}}{2} M_c \varepsilon_\theta{}^{ep}\left(\frac{R_p}{r}\right)^2 + M_c \varepsilon_\theta{}^{ep}\right] \end{cases} \tag{25}$$

3.2.2. Displacement of Plastic Softening Zone

According to the flow law and geometric Equation (3) in the plastic softening zone, the following results can be obtained:

$$\frac{\partial u_p}{\partial r} + \eta_1 \frac{u_p}{r} = 0 \tag{26}$$

By solving the differential Equation (26), the following result is obtained

$$u_p = C_3 r^{-\eta_1} \tag{27}$$

Since the boundary condition of the crushed zone is $r = R_p$ and $u_p = u_e$, it can be deduced that:

$$C_3 = NR_p{}^{\eta_1+1} \tag{28}$$

where: $N = \frac{1+\mu}{E}\left[(1+\lambda)(1-\mu)p_0 - \sigma_{ep} + 2\cos\theta(1-\lambda)(1-\mu)p_0\right]$.

By substituting Equation (28) into Equation (27), the displacement field of the plastic softening zone can be obtained:

$$u_p = Nr^{-\eta_1} R_p{}^{\eta_1+1} \tag{29}$$

*3.3. Mechanical Analysis of Crushed Zone*

3.3.1. Stress of Fracture Zone

The surrounding rock in the crushed zone not only meets the yield criterion of Equation (1), but also the equilibrium differential Equation (2). The following results can be obtained by combining the two equations:

$$\sigma_r{}^b = \frac{L_b}{1-K_b} + C_4 r^{K_b-1} \tag{30}$$

where: $K_b = \frac{1+\sin\varphi_b}{1-\sin\varphi_b}$, $L_b = \frac{2c_b \cos\varphi_b}{1-\sin\varphi_b}$.

With the stress boundary condition being $r = R_0$, $\sigma_r{}^b = p_i$, it can be deduced that:

$$C_4 = \left(p_i - \frac{L_b}{1-K_b}\right) R_0{}^{1-K_b} \tag{31}$$

The stress state of the crushed zone can be obtained from Equations (1), (30) and (31):

$$\begin{cases} \sigma_r{}^b = \frac{L_b}{1-K_b} + \left(p_i - \frac{L_b}{1-K_b}\right)\left(\frac{R_0}{r}\right)^{1-K_b} \\ \sigma_\theta{}^b = \frac{L_b}{1-K_b} + K_b\left(p_i - \frac{L_b}{1-K_b}\right)\left(\frac{R_0}{r}\right)^{1-K_b} \end{cases} \tag{32}$$

3.3.2. Displacement of Crushed Zone

According to the plastic flow law and the geometric equation, the following conclusions can be obtained:

$$\frac{\partial u_b}{\partial r} + \eta_2 \frac{u_b}{r} = 0 \tag{33}$$

By solving the differential equation, the result can be obtained

$$u_b = C_5 r^{-\eta_2} \tag{34}$$

With the boundary condition being $r = R_b$ and $u_b = u_p$, it can be deduced that:

$$C_5 = N R_p{}^{\eta_1+1} R_b{}^{\eta_2-\eta_1} \tag{35}$$

By substituting Equation (35) into Equation (34), the displacement of the crushed zone can be obtained:

$$u_b = N r^{-\eta_2} R_p{}^{\eta_1+1} R_b{}^{\eta_2-\eta_1} \tag{36}$$

**4. Analysis of the Range of Plastic and Crushed Zone**

According to the strain softening model, the strength of surrounding rock softens to the residual value at the interface between the plastic zone and the crushed zone. Equation of softening modulus is as follows:

$$M_c \left( \varepsilon_\theta{}^{pb} - \varepsilon_\theta{}^{ep} \right) = c_0 - c_b \tag{37}$$

On the elastic-plastic interface, it is known from Equation (29): $\varepsilon_\theta{}^{ep} = N$.

On the interface between plastic softening zone and fracture zone, when $r = R_b$ is substituted into Equation (29), the result is as follows:

$$u_p = N R_b{}^{-\eta_1} R_p{}^{\eta_1+1} \tag{38}$$

Then:

$$\varepsilon_\theta{}^{pb} = \frac{u_p}{R_b} = N \left( \frac{R_p}{R_b} \right)^{\eta_1+1} \tag{39}$$

By combining Equations (37)–(39), we can get:

$$\frac{R_p}{R_b} = \left( \frac{c_0 - c_b}{N M_c} + 1 \right)^{\frac{1}{\eta_1+1}} = \xi^{\frac{1}{\eta_1+1}} \tag{40}$$

At the interface of plastic softening zone and fracture zone, the stress continuity condition are $r = R_b$ and $\sigma_r{}^p = \sigma_r{}^b$. Substituting them into Equations (25) and (32), we can get:

$$\left[ \sigma_{ep} - m(A+B) \right] \left( \frac{R_p}{R_b} \right)^{1-n} + m \left[ A + B \left( \frac{R_p}{R_b} \right)^2 \right] = \frac{L_b}{1-K_b} + \left( p_i - \frac{L_b}{1-K_b} \right) \left( \frac{R_0}{R_b} \right)^{1-K_b} \tag{41}$$

By solving the above Equation (41), it can be known that:

$$R_b = R_0 \left[ \frac{(1-K_b)(\sigma_{ep} - mA - mB)\xi^{\frac{1-n}{1+\eta_1}} + m(1-K_b)\left( A + B\xi^{\frac{2}{\eta_1+1}} \right) - L_b}{(1-K_b)p_i - L_b} \right]^{\frac{1}{K_b-1}} \tag{42}$$

By combining Equations (40) and (42), the following can be obtained:

$$R_p = \xi^{\frac{1}{1+\eta_1}} R_0 \left[ \frac{(1-K_b)(\sigma_{ep} - mA - mB)\xi^{\frac{1-n}{1+\eta_1}} + m(1-K_b)\left(A + B\xi^{\frac{2}{\eta_1+1}}\right) - L_b}{(1-K_b)p_i - L_b} \right]^{\frac{1}{K_b-1}} \tag{43}$$

## 5. Sensitivity Analysis of Influencing Factors

The extent of EDZ directly affects the deformation control and support parameter design of roadway [24]. Equation (43) shows that the extent of EDZ is mainly affected by the original rock stress, lateral pressure coefficient, cohesion, internal friction angle and support resistance. In order to analyze the influence of various factors, the single factor sensitivity analysis method is adopted here. The calculated results are compared with the classical Rubin's solution [25] to verify the applicability of our proposed model.

The Rubin's solution is as follows [25]:

$$R_p = R_0 \left\{ \frac{[p_0(1+\lambda) + 2c\cot\varphi](1 - \sin\varphi)}{2p_i + 2c\cot\varphi} \right\}^{\frac{1-\sin\varphi}{2\sin\varphi}} \times \left\{ 1 + \frac{p_0(1-\lambda)(1-\sin\varphi)\cos 2\theta}{[p_0(1+\lambda) + 2c\cot\varphi]\sin\varphi} \right\} \tag{44}$$

Here, the original rock stress ($p_0$) takes the value of 20 MPa, the cohesion ($c$) is 2 MPa, the internal friction angle ($\varphi$) is 30°, the support resistance ($p_i$) is 0.2 MPa. The calculated results show that the predictions from our model are slightly larger than those of the Rubin's solution. This can be justified by the fact that the factors such as strain softening and dilatancy are further considered in our model. In addition, we use a three-zone composite mechanical model while in the Rubin's solution all the surrounding rock are assumed as ideal elastic-plastic bodies. This indicates that our results are more accurate while Rubin's solution will underestimate the extent of EDZ. The detailed single factor sensitivity analysis is presented as below:

### 5.1. Lateral Pressure Coefficient

When the lateral pressure coefficients are 0.5, 1, 1.5, 2, and 2.5, respectively, and other parameters remain unchanged, the Rubin's solution and the solution in this paper are compared and shown in Figure 5.

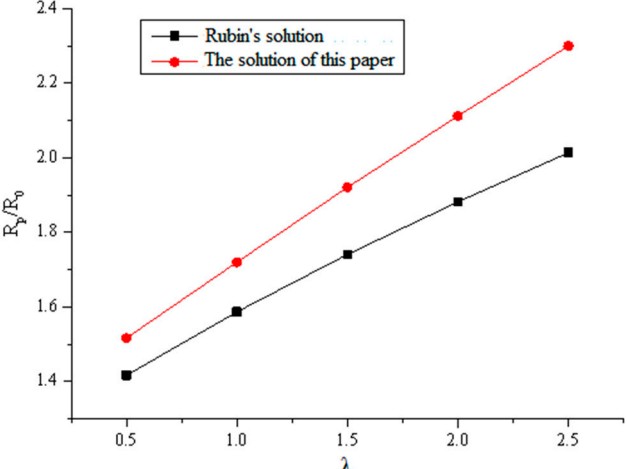

**Figure 5.** Plastic range under different lateral pressure coefficient.

It can be seen from Figure 5 that the size of plastic zone is directly proportional to the lateral pressure coefficient. The calculated result of plastic zone in this paper is slightly larger than that of the Rubin's solution. When the lateral pressure coefficient exceeds 2,

the influence of the lateral pressure coefficient becomes more remarkable. Therefore, the deeper the roadway is buried, the greater the lateral pressure coefficient, and the larger the extent of EDZ.

*5.2. Original Rock Stress*

When the original rock stress is set as 5, 10, 15, 20, and 25, respectively, the lateral pressure coefficient is taken as 2, and other parameters remain unchanged, the Rubin's solution of plastic zone distribution extent and the solution in this paper are compared and shown in Figure 6.

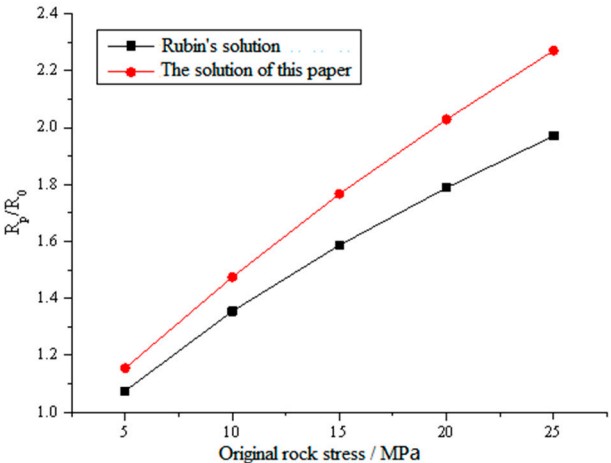

**Figure 6.** Plastic range under different original rock stress.

It can be seen from Figure 6 that with the increase of the original rock stress, the extent of the plastic zone increases significantly. This is because the shallower surrounding rock releases the stress by fractures and plastic deformation, and the stress concentration zone is transferred to the deeper regions, resulting in the increase of plastic zone range. This can also justify why the superposed mining pressure will increase the plastic zone extent and the difficulty of roadway support.

*5.3. Cohesion*

When the cohesion is taken as 2, 3, 4, 5, and 6, respectively, and the lateral pressure coefficient is taken as 2, and other parameters remain unchanged, the Rubin's solution of plastic zone distribution range and the solution in this paper are compared and shown in Figure 7.

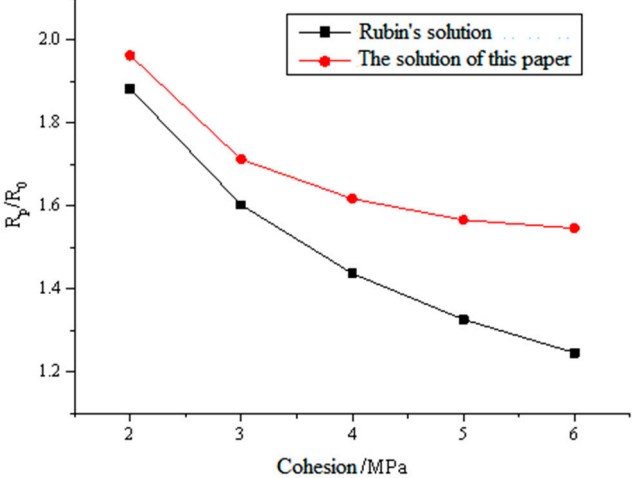

**Figure 7.** Plastic range under different cohesions.

It can be seen from Figure 7 that the range of plastic zone decreases significantly with the increase of cohesion. This is because the increase of cohesion leads to the increased strength of surrounding rock, and improved ability of rock mass to resist deformation. In this case the range of plastic zone can be obviously contained. The integrity of rock mass can be improved by means of grouting or injection anchor, so as to increase the cohesion of rock mass and reduce the extent of plastic zone.

### 5.4. Support Resistance

When the support resistance is set as 0.2, 0.6, 1.0, 1.4, and 1.8, respectively, the lateral pressure coefficient is taken as 9, and other parameters remain unchanged, the Rubin's solution of plastic zone distribution range and the solution in this paper are compared and shown in Figure 8.

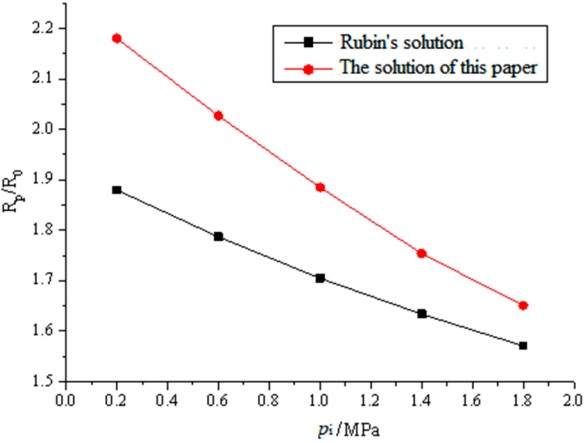

**Figure 8.** Plastic range under different support resistances.

It can be seen from Figure 8 that the support resistance of roadway also plays an important role in reducing the extent of the surrounding rock plastic zone. However, the support resistance provided by the support methods such as anchor bolt is very limited, generally not more than 0.3 MPa. Therefore, for the large deformation roadway, unless the bolt support strength is improved noticeably, the control effect of roadway deformation would be rather poor. Accordingly, the comprehensive support method such as bolt plus grouting should be adopted.

### 6. Case Study

Our established theoretical model is used to analyzed the EDZ development of a roadway in a coal mine locating in Pingdingshan, Henan Province, China, subjected to the influence of mining pressures. The extent of EDZ before and after mining was calculated. Since the original cross section of this roadway was a straight-wall semicircular arch, the radius of roadway was firstly corrected by the circumscribed circle radius method:

$$r^* = \left[ (2h + B) + \frac{B^2}{2h + B} \right] / 4 \tag{45}$$

where $r^*$ denotes the equivalent radius of roadway; $h$ is the height of the straight wall; and $B$ stands for the net width. In this case, $h = 1.4$ m and $B = 4.2$ m, give $r^* = 2.38$ m. The Table 1 below gives the mechanical parameters of the surrounding rock.

**Table 1.** Mechanical parameters of roadway surrounding rock.

| $E$ (GPa) | $\mu$ | $c_0$ (MPa) | $c_b$ (MPa) | $\varphi_0$ (MPa) | $\varphi_b$ (MPa) | $p_0$ (MPa) | $p_i$ (MPa) |
|---|---|---|---|---|---|---|---|
| 7.5 | 0.25 | 1.52 | 0.5 | 30 | 24 | 15 | 0.2 |

After mining, the stope line was 100 m away from the roadway, and the roadway was subjected to the supporting pressure of the working face, which is about 5 MPa. In addition, the original stress of surrounding rock increased from 15 MPa to 20 MPa after mining. These parameters were substituted into the Equations (29), (36), (42), and (43), and the results were shown in Table 2, which shows that after mining, the depth of crushed zone increases from 1.1 m to 1.39 m, and the depth of plastic zone increases from 1.73 m to 2.08 m. Therefore, the extent of EDZ further increases after mining, which will significantly shorten the effective anchoring section of the anchor bolt and reduced its anchoring force, and increase the deformation of surrounding rock.

**Table 2.** The extent and deformation of plastic zone and crushed zone before and after mining.

| | Crushed Zone | | Plastic Zone | |
| --- | --- | --- | --- | --- |
| | **Depth (m)** | **Displacement (mm)** | **Depth (m)** | **Displacement (mm)** |
| Before mining | 1.10 | 146 | 1.73 | 38 |
| After mining | 1.39 | 483 | 2.08 | 82 |

## 7. Conclusions

Based on the elastic-plastic mechanics, an analytical model for the excavation damage zone in the tunnel surrounding rock is proposed, which considers the combined effects of the strain softening and dilatancy during the rock deformation. The tunnel surrounding rock is divided into three zones, including the elastic zone, plastic zone, and crushed zone. The corresponding analytical expressions of stress state, displacement field, and extant of EDZ in the tunnel surrounding rock were derived. The comparison and analysis with classic Rubin's solution verify the correctness of the theoretical model. By performing a single factor sensitivity analysis, the influence of different factors such as lateral pressure coefficient, in-situ stress, cohesion, and support resistance was analyzed. The proposed model was further used to evaluate the EDZ development of a roadway subjected to the mining influence. The calculation results show that the maximum displacement of the roadway surface increases from 146 mm to 483 mm, and the depth of plastic zone increases from 1.73 m to 2.08 m The proposed model can not only benefit for the future numerical and experimental studies, but also the extensive engineering practices.

**Author Contributions:** Conceptualization, X.X.; methodology, X.X.; software, C.Z. (Chun Zhu); validation, X.X., C.Z. (Chun Zhu) and Y.Z.; formal analysis, C.Z.; investigation, C.Z. (Chun Zhu); resources, X.X.; data curation, C.Z. (Chun Zhu); writing—original draft preparation, X.X.; writing—review and editing, C.Z. (Chunlin Zeng); visualization, C.Z. and C.G.; supervision, Y.Z.; project administration, Y.Z.; funding acquisition, Y.Z. All authors have read and agreed to the published version of the manuscript.

**Funding:** This work is supported by the National Natural Science Foundation of China (No. 52061135111), the National Natural Science Foundation of China (No. 51874289), and the China Postdoctoral Science Foundation (No. 2020T130702).

**Informed Consent Statement:** Informed consent was obtained from all subjects involved in the study.

**Data Availability Statement:** The manuscript data used to support the findings of this study are available from the corresponding author upon request.

**Conflicts of Interest:** The authors declare no conflict of interest. The funders had no role in the design of the study; in the collection, analyses, or interpretation of data; in the writing of the manuscript; or in the decision to publish the results.

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
