# Peer review of "An Analytical Model for the Excavation Damage Zone in Tunnel Surrounding Rock"

_minerals, doi:10.3390/min12101321_

Round 1

Reviewer 1 Report

Based on elastic-plastic theory, an analytical solution to the stress and deformation field of the excavation damage zone in the tunnel surrounding rock was derived in this manuscript. The effect of different factors on the model were quantified and analyzed. The findings from this manuscript can provide valuable insights for the design and safety assessment in the mining engineering. The following issues/questions should be addressed properly prior to publication.

(1) The Introduction part is not convincing enough to introduce the study. A more comprehensive and critical literature survey should be provided to identify the knowledge gap.

(2) Is the n in equation (42) or (43) representing η? Typos?

(3) Line 274, the corresponding symbols should be appended. For example, the original rock stress (p0).

(4) Please clarify the origin of Rubin’s solution.

(5) It is suggested that adding the calculation of the engineering case to the article.

(6) The conclusion part should not be the repetition of the abstract. Please improve it.

(7) Unify the terminology in this paper. For example, what is the relation between the excavation damage zones and three zones defined in the Section 2?

(8) Supplement proper citations for those commonly used equations in Section 2.

Author Response

Thanks for the comments from the reviewer. We did our best to modify this article based on your valuable comments and the specific modification details are as follows:

(1) The Introduction part is not convincing enough to introduce the study. A more comprehensive and critical literature survey should be provided to identify the knowledge gap.

We have further modified our Introduction part in the revised manuscript.

(2) Is the n in equation (42) or (43) representing η? Typos?

Yes. This typo has been corrected in the revised manuscript.

(3) Line 274, the corresponding symbols should be appended. For example, the original rock stress (p0).

The corresponding symbols have been added to the revised manuscript.

(4) Please clarify the origin of Rubin’s solution.

The corresponding reference has been added to the revised manuscript.

(5) It is suggested that adding the calculation of the engineering case to the article.

An engineering case has been added to the revised manuscript.

(6) The conclusion part should not be the repetition of the abstract. Please improve it.

We have further modified our Conclusion part in the revised manuscript.

(7) Unify the terminology in this paper. For example, what is the relation between the excavation damage zones and three zones defined in the Section 2?

The range of the excavation damage zone includes the crushed zone and plastic softened zone. We have added this definition in the Section 2.

(8) Supplement proper citations for those commonly used equations in Section 2.

The corresponding references have been cited in the revised manuscript.

Reviewer 2 Report

The paper addresses the problem of predicting the extent of the damage zones around underground excavations in rock. An analytical solution is developed for the case of circular tunnels, assuming a 2D model. It is based on elasto-plastic constitutive relations taking into account softening and dilatant behavior. The results of parametric studies are presented to investigate the effects of initial rock stresses and rock strength parameters. The proposed solution provides a simple tool to estimate the tunnel excavation effects and can also be helpful in the verification of numerical models based on similar constitutive assumptions. Some revisions are proposed:

1) Include references to the codes cited in the Introduction.

2) Section 2. Verify the format of equations and of the symbols in the text.

3) Section 5. State the assumptions implied in Rubin’s solution, so that the reasons for the differences between the two models can be clearly understood.

Author Response

Thanks for the comments from the reviewer. We did our best to modify this article based on your valuable comments and the specific modification details are as follows:

1) Include references to the codes cited in the Introduction.

The corresponding references for the mentioned codes and software, e.g., FLAC3D, RFPA2D, and LS-DYNA, have been added to the introduction section.

2) Section 2. Verify the format of equations and of the symbols in the text.

The format of equations and symbols in the manuscript has been modified according to the journal’s requirements.

3) Section 5. State the assumptions implied in Rubin’s solution, so that the reasons for the differences between the two models can be clearly understood.

Rubin’s solution assumed that all the surrounding rock are ideal elastic-plastic bodies, while a three-zone composite mechanical model is adopted in our model. That is why the difference occurs.

The corresponding explanations have been added to Section 5 in the revised manuscript.